# The Sustainable Production of Terpenoids in Cyanobacterial Chassis

**DOI:** 10.3390/microorganisms13061342

**Published:** 2025-06-10

**Authors:** Bo Hong, Ling Qiu, Ruo Lv, Zongxia Yu

**Affiliations:** 1Jiangxi Key Laboratory for Sustainable Utilization of Chinese Materia Medica Resources, Lushan Botanical Garden, Chinese Academy of Sciences, Jiujiang 332900, China; hongb@lsbg.cn (B.H.); 13763962814@163.com (L.Q.); lr18779230313@163.com (R.L.); 2College of Life Science, Nanchang University, Nanchang 330031, China

**Keywords:** terpenoids, cyanobacteria, metabolic engineering, synthetic biology

## Abstract

Terpenoids, which are widely distributed in nature, possess diverse biological activities, physiological functions, and economic values. They are extensively exploited by plants and microorganisms. However, the abundance of terpenoids in natural hosts is extremely low, making it difficult to meet the market demands. In recent years, along with the advancement of metabolic engineering and synthetic biology, it has become feasible for microorganisms to produce exogenous terpenoids sustainably. Cyanobacteria and other photosynthetic microorganisms have attracted growing attention due to their capacity to produce terpenoids by harnessing light and carbon dioxide. In this article, we comprehensively summarize the biosynthetic pathways of terpenoids and the progress in utilizing cyanobacteria as chassis for the production of terpenoids, and further discuss strategies for augmenting the yields of terpenoids.

## 1. Introduction

Terpenoids are one of the major secondary metabolites and possess the most abundant compounds in nature. Terpenoids are synthesized with 2-methyl-1,3-butadiene (also known as isoprene) as the fundamental unit under the catalysis of a series of terpenoid synthases and modifying enzymes [1]. They are prevalently found in plants, insects, microorganisms, and some marine organisms and feature many types and diverse structures [2,3]. During the synthesis process, the carbon skeleton can undergo modifications, such as cyclization, carbon atom addition, and reduction. Depending on the number of basic units of isoprene and different connection patterns, terpenoids can be classified into hemiterpenoids (C5), monoterpenoids (C10), sesquiterpenoids (C15), diterpenoids (C20), triterpenoids (C30), tetraterpenoids (C40), or polyterpenoids (C > 40) [1]. The majority of terpenoids possess specific physiological functions and high application values, thus they have been extensively developed and utilized in the fields of medicine, flavors, food, and biofuels [1,4]. For instance, the monoterpenoid geraniol is widely employed as an ingredient in flavorings and cosmetics [5]. The sesquiterpenoids farnesene and bisabolene can be utilized as biofuels [1,6]. The diterpenoid taxol is an important clinical anticancer drug [7]. Squalene, which serves as a moisturizer in cosmetics, is a triterpenoid compound [8]. The tetriterpenoids astaxanthin and lycopene possess antioxidant, anti-inflammatory, and anticancer effects and are frequently used in nutritional health products [9,10].

Terpenoids play a crucial role in human industry, life, and health. However, their natural amounts in their original organisms are minimal, and the extraction process is complex and time-consuming, which seriously hampers the large-scale production of terpenoids. For instance, fully mature *Taxus brevifolia* can yield only approximately 260 mg of taxol through the traditional extraction methods. To obtain a sufficient dose of taxol for treating a single cancer patient, the material from 2–4 mature trees is required [11,12]. Consequently, the extraction of terpenoids from natural sources is not only limited in supply, but also results in ecological damage. Although chemical synthesis has been employed for terpenoid production, flaws, such as it having a complex process, multiple steps, low yields, and causing environmental pollution, seriously restrict its application [11]. Recent advances in synthetic biology have facilitated the construction of microbial cell factories for heterologous terpenoid production, offering a sustainable solution to the current production problems. This approach shows significant promise for supplying high-value terpenoids to various industries, including pharmaceutical, fragrance, and biofuel, while addressing the key challenges of the conventional production methods [7].

The biosynthesis of terpenoids primarily consists of the construction and optimization of metabolic pathways in diverse organisms to hyperproduce terpenoids through metabolic engineering or genetic modification. Currently, model microorganisms, such as *Escherichia coli* and *Saccharomyces cerevisiae,* are the most extensively studied and applied chassis due to their advantages, including well-characterized genomes, ease of manipulation, and suitability for large-scale production [13]. However, these microorganisms consume a lot of energy during fermentation and generate substantial amounts of CO_2_, which contradicts with sustainable production. In contrast, cyanobacteria, which are photosynthetic autotrophic prokaryotes, can convert light energy and CO_2_ into valuable products without relying on organic carbon sources [14,15]. Cyanobacteria naturally produce a variety of photosynthetic pigments, such as carotenoids and chlorophyll, which are tetraterpenes, implying they contain an abundant isoprene pool and making them ideal candidates for terpenoid production [16]. Therefore, when compared to *E. coli* and *S. cerevisiae*, cyanobacteria offer distinct advantages as microbial cell factories, including less energy consumption, a low cost, high energy efficiency, and abundant endogenous precursors. Furthermore, model cyanobacterial species, such as *Synechocystis* sp. *PCC* 6803 (hereafter *Synechocystis* 6803) and *Synechococcus elongatus PCC* 7942 (hereafter *Synechococcus* 7942), exhibit rapid growth, simple genetic manipulation, a limited range of metabolites, a plasma-like membrane system, and pre-existing terpenoid synthesis pathways, making them suitable for being engineered into terpenoid synthesis chassis [17,18,19,20,21].

We systematically reviewed and analyzed the recent studies on terpenoid biosynthesis in various cyanobacterial hosts. These studies encompass enzyme function validation, quantitative yield data, fermentation system optimization, and recent breakthroughs. In this study, we summarize the current findings and provide insights into future prospects. We first introduce the background of terpenoid biosynthesis, which is composed of upstream and downstream modules. We then focus on the progress in metabolic engineering for terpenoid production in cyanobacteria. Furthermore, we present the synthesis strategies for several key terpenoids in cyanobacteria, providing references for the rational design of cyanobacterial chassis. This work offers new perspectives for the green, efficient, and sustainable production of terpenoids and promotes the application of cyanobacterial microcell factories in terpenoid biosynthesis.

## 2. Synthetic Pathways of Terpenoids

To successfully engineer terpenoid biosynthesis in cyanobacteria, a comprehensive understanding of the terpenoid synthesis pathway is imperative. The terpenoid biosynthetic pathway can be conceptually divided into two distinct modules, the upstream module, responsible for the synthesis of the essential precursor molecules isopentenyl pyrophosphate (IPP) and dimethylallyl pyrophosphate (DMAPP), and the downstream module, where a series of isoprene synthases and terpenoid synthases utilize IPP and DMAPP as substrates to generate a wide array of terpenoid compounds.

### 2.1. The Upstream Module for the Synthesis of IPP and DMAPP

Terpenoids, despite their structural complexity and diversity, are primarily derived from two fundamental C5 units: IPP and its isomer DMAPP. In nature, the synthesis of IPP and DMAPP mainly occurs along two pathways: the methylerythritol phosphate (MEP) pathway and the mevalonate (MVA) pathway (Figure 1).

The MEP pathway is predominantly found in prokaryotes and catalyzed by seven enzymes in plastids. Firstly, glyceraldehyde-3-phosphate (G3P) from the Calvin cycle is condensed with pyruvate from glycolysis, catalyzed by DXP synthase (DXS) with thiamine pyrophosphate (TPP) as a cofactor, to form 1-deoxy-d-xylulose-5-phosphate (DXP). DXP is then reduced to MEP by DXP reductoisomerase (DXR) using NADPH as the reducing agent. Subsequently, 2-C-methyl-D-erythritol-4-phosphate cytidyltransferase (IspD) catalyzes the formation of 4-diphosphocytidyl-2-C-methyl-D-erythritol (CDP-ME), which is then phosphorylated by CDP-ME kinase (IspE) in the presence of ATP to yield 4-diphosphocytidyl-2-C-methyl-d-erythrose-2-phosphate (CDP-MEP). CDP-MEP is subsequently processed by 2-C-methyl-erythritol-2,4-cyclodiphosphate (MEcPP) synthase (IspF), which catalyzes the cyclization and cleavage of CDP-MEP to form MEcPP. Finally, MEcPP undergoes reduction by (E)-4-hydroxy-3-methyl-but-2-enyl diphosphate (HMBPP) synthase (IspG) to produce HMBPP, which is then further reduced to IPP and DMAPP by HMBPP reductase (IspH). Additionally, IPP and DMAPP can be interconverted through the action of the isopentenyl diphosphate isomerase (IDI) [22,23,24].

**Figure 1 microorganisms-13-01342-f001:**
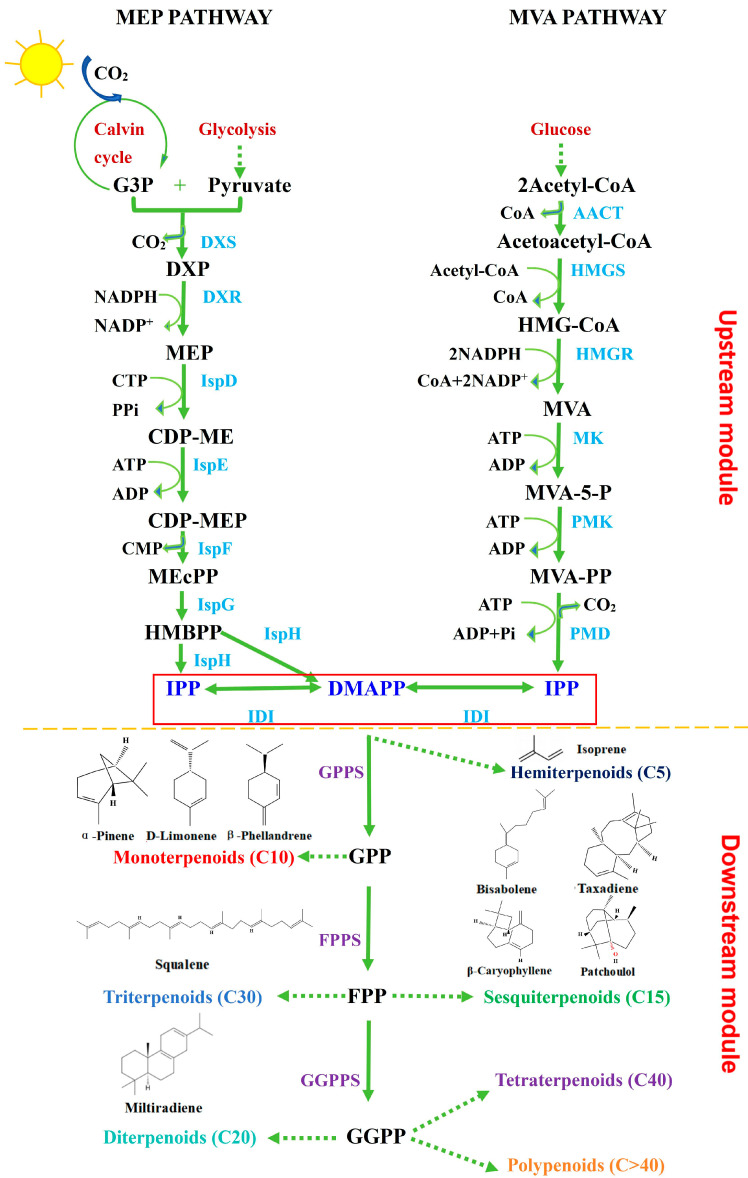
Synthetic pathways of terpenoids. G3P: glyceraldehyde-3-phosphate; DXP: 1-deoxy-d-xylulose-5-phosphate; DXS: DXP synthase; DXR: DXP reductoisomerase; MEP: methylerythritol phosphate; IspD: 2-C-methyl-D-erythritol 4-phosphate cytidyltransferase; CDP-ME: 4-diphosphocytidyl-2-C-methyl-d-erythritol; IspE: CDP-ME kinase; CDP-MEP: 4-diphosphocytidyl-2-C-methyl-d-erythrose-2-phosphate; MEcPP: 2-C-methyl-erythritol-2,4-cyclodiphosphate; HMBPP: (E)-4-hydroxy-3-methyl-but-2-enyl diphosphate; IspG: HMBPP synthase; IspH: HMBPP reductase; AACT: Acetyl-CoA C-acetyl transferase; HMGS: 3-hydroxy-3-methyl-glutaryl-CoA synthase; HMG-CoA: 3-hydroxy-3-methyl-glutaryl-CoA; HMGR: HMG-CoA reductase; MVA: mevalonate; MVA-5-P: mevalonate-5-phosphate; MK: mevalonate kinase; MVA-PP: mevalonate-5-diphosphate; PMK: mevalonate-5-phosphate kinase; PMD: mevalonate-5-diphosphate decarboxylase; IPP: isoprene pyrophosphate; DMAPP: dimethylallyl pyrophosphate; IDI: isopentenyl diphosphate isomerase; GPPS: geranyl pyrophosphate synthase; GPP: geranyl pyrophosphate; FPP: farnesyl pyrophosphate; FPPS: FPP synthase; GGPP: geranylgeranyl pyrophosphate; GGPPS: GGPP synthase. Solid red box represents component that is common to both MEP and MVA pathways. Schematic adapted from [25].

The MVA pathway exists primarily in eukaryotes and some bacteria. It contains six enzymatic reactions in the cytoplasm. Initially, acetyl-CoA acetyltransferase (AACT) and 3-hydroxy-3-methylglutaryl-CoA synthase (HMGS) catalyze the condensation of acetyl-CoA to form acetoacetyl-CoA and 3-hydroxy-3-methylglutaryl-CoA (HMG-CoA). The next critical step involves the reduction of HMG-CoA to MVA by HMG-CoA reductase (HMGR), a rate-limiting enzyme in the pathway, which consumes two molecules of NADPH. Subsequently, MVA is phosphorylated to mevalonate-5-phosphate (MVA-5-P) by mevalonate kinase (MK), and MVA-5-P is further phosphorylated to mevalonate-5-diphosphate (MVA-PP) by mevalonate-5-phosphate kinase (PMK), consuming two ATP molecules in the process. Finally, mevalonate-5-diphosphate decarboxylase (PMD) decarboxylates MVA-PP to generate IPP, consuming one ATP molecule. Once IPP is produced, DMAPP can be synthesized from IPP via isomerization by IDI [23,24].

### 2.2. The Downstream Module for the Synthesis of Various Terpenoids

The C5 precursor IPP and its isomer DMAPP are the building blocks of terpenoids, and they are further catalyzed to form various types of terpenoid precursor by distinct isoprene phosphate synthases as follows: Geranyl pyrophosphate synthase (GPPS) catalyzes the formation of geranyl pyrophosphate (GPP) from IPP and DMAPP, serving as a precursor for monoterpenoids. Farnesyl pyrophosphate synthase (FPPS) further synthesizes farnesyl pyrophosphate (FPP), the precursor for sesquiterpenoids, by using IPP and GPP as substrates. Geranylgeranyl pyrophosphate synthase (GGPPS) then combines IPP with FPP to produce geranylgeranyl pyrophosphate (GGPP), a precursor for diterpenoids. The precursors for triterpenoids and tetraterpenoids are formed from the direct condensation of two IPPs or two GGPPs, respectively [25]. These precursors undergo cyclization, dephosphorylation, hydroxylation, or glycosylation catalyzed by various terpenoid synthases, cytochrome P450, or glycosyltransferase to yield diverse terpenoid derivatives (Figure 1).

## 3. Enhance Terpenoids Production via Metabolic Engineering of Cyanobacteria

The biosynthesis of terpenoids requires substrates such as acetyl-CoA and pyruvate; the C5 units from the upstream MEP and MVA pathways; and precursors like GPP, FPP, and GGPP from the downstream module. Photosynthetic cyanobacteria are capable of producing the substrates through the utilization of light and CO_2_ and possess both the upstream MEP pathway and the downstream terpenoid biosynthetic pathway. These provide the necessary conditions for terpenoid synthesis. However, due to complex regulatory mechanisms, metabolic flux toward terpenoid production can be inefficient, which limits the biosynthesis of certain terpenoids. To overcome these limitations and increase the yield of the target terpenoids, metabolic engineering is usually employed. These approaches involve the introduction of the terpenoid synthesis pathway into cyanobacteria through transgenesis, as well as the identification and modification of key enzymes to enhance the supply of precursors and tune the metabolic flux toward the desired terpenoid products. Table 1 summarizes the recent studies on terpenoid biosynthesis in cyanobacteria.

### 3.1. Metabolic Engineering of the MEP Pathway

The MEP pathway is the main metabolic route that directs pyruvate and G3P in terpenoid biosynthesis in many cyanobacteria, providing the essential C5 units of IPP and DMAPP for the subsequent reactions. Therefore, strengthening the MEP pathway not only redirects the flow of pyruvate and G3P towards terpenoid synthesis, but also increases the availability of the precursors required for terpenoid production. The strategies to improve the MEP pathway in cyanobacteria typically focus on optimizing and over-expressing key rate-limiting enzymes. DXS is the first rate-limiting enzyme in the MEP pathway. The engineering of DXS can enhance the utilization of pyruvate and G3P to produce more DMAPP and IPP for terpenoid synthesis. For instance, the over-expression of *dxs* from *Coleus forskohlii* in *Synechocystis* 6803 led to a 4.2-fold increase in the titer of the forskolin precursor 13R-manoyl oxide (13R-MO) [32]. Similarly, the over-expression of *dxs* derived from *Botryococcus braunii* in the limonene-producing strain *Synechococcus* 7942 resulted in an increase in limonene productivity, from 65.4 μg/L/OD/d to 76.3 μg/L/OD/d [33]. In contrast, the over-expression of the endogenous *dxs* gene in *Synechococcus* 7942 only resulted in a 20% increase in isoprene production [34]. To further improve the yield, researchers often over-express both the DXS and IDI enzymes. For example, the simultaneous over-expression of *dxs* and *idi* from *Schizonepeta tenuifolia* in *Synechocystis* 6803 resulted in a 1.4-fold increase in limonene production, from 41 μg L^−1^ day^−1^ to 56 μg L^−1^ day^−1^ [35]. Likewise, Rodrigues and Lindberg [29] demonstrated that the co-over-expression of *dxs* from *C. forskohlii* and *idi* from *Synechocystis* in *Synechocystis* 6803 led to nearly a two-fold increase in the production of (E)-α-bisabolene. However, optimizing DXS does not always lead to enhanced terpenoid production. For example, Shimada et al. [36] found that the over-expression of endogenous *dxs* in recombinant *Synechocystis* 6803 did not significantly alter the carotenoid levels under light conditions. Moreover, DXS over-expression inhibited the accumulation of echinenone and ketomyxol fucoside under dark conditions, indicating the context-dependent effects of metabolic engineering in cyanobacteria.

The aforementioned studies emphasize that DXS is a critical rate-limiting enzyme in the MEP pathway, yet it causes CO_2_ loss during DXP formation (Figure 1). Exploring strategies to bypass DXS and enhance MEP flux is an intriguing research topic. A recent study by Zhou et al. [27] engineered the pentose phosphate pathway in cyanobacteria to directly synthesize DXP from ribulose-5-phosphate (Ru5P), thus circumventing DXS. They achieved this by introducing mutations in the *aceE* gene (encoding the E1 subunit of pyruvate dehydrogenase) and the *ribB* gene (encoding a subunit of the pyruvate dehydrogenase complex) into *Synechococcus* 7942 to compensate for the deletion of DXS, which led to a significant increase in the production rate of isopentenol [27]. In addition to DXS and IDI, other key enzymes in the MEP pathway have also been identified. For instance, Gao et al. [34] demonstrated that over-expressing the *ispG* gene from *Thermosynechococcus elongatus* in *Synechococcus* 7942 resulted in a 60% increase in isoprene production. Zhou et al. [27] further confirmed this finding by over-expressing both the *ispG* and *ispH* genes in cyanobacterial chassis synthesizing DXP via the Ru5P pathway, and this led to an accumulation of 105.2 mg/L of isopentenol.

These findings demonstrate that strengthening the MEP pathway in cyanobacteria can be achieved not only through the over-expression of key enzymes, but also by redesigning the pathway to minimize carbon loss. However, Rodrigues and Lindberg [29] have pointed out that in the absence of a carbon sink, the over-expression of DXS and IDI can cause significant growth impairment. Therefore, a broader array of strategies should be explored to boost the synthesis of terpenoids.

### 3.2. Introduction of Exogenous MVA

During the evolution of cyanobacteria, certain species lost the native MVA pathway and retained only the MEP pathway [37]. Consequently, the heterologous expression of the MVA pathway in cyanobacteria has emerged as a promising strategy for terpenoid biosynthesis, which has been widely applied to *E. coli.* Recently, there is growing evidence supporting that introducing the MVA pathway can substantially augment the terpenoid yields in cyanobacteria. For example, Formighieri and Melis [38] successfully reconstituted the MVA pathway in *Synechocystis* 6803 by expressing the *atoB* gene from *E. coli*; *hmgS* from *Enterococcus faecalis*; *hmgR* from *Enterococcus faecalis*; as well as the *mk*, *pmk*, *pmd*, and *idi* genes from *Streptococcus pneumoniae*, resulting in a marked increase in terpenoid production. Similarly, the co-expression of the MVA pathway along with GPPS and β-phellandrene synthase (PHLS) in *Synechocystis* 6803 further elevated the yield of β-phellandrene [18]. These findings underline the feasibility of heterologously expressing the MVA pathway in cyanobacteria to boost terpenoid biosynthesis. Moreover, the combined expression of both the MVA and MEP pathways holds potential for further improving the terpenoid yields in cyanobacteria.

### 3.3. Optimization and Regulation of Terpenoid Synthases

Cyanobacteria inherently synthesize a variety of photosynthetic pigments like carotenoids and chlorophyll, which consume a significant portion of the GGPP pool, thus diminishing its availability for the production of other terpenoids. To overcome this limitation, the optimization or reconstruction of the downstream biosynthetic pathways for terpenoid production in cyanobacteria is essential. Enhancing the biosynthesis of terpenoids can be achieved through the optimization and over-expression of terpenoid synthases. For instance, Wang et al. [33] over-expressed a codon-optimized, signal peptide-truncated limonene synthase from *Mentha spicata* in *Synechococcus* 7942, achieving a limonene production rate of 8.5 µg/L/OD/day. Similarly, Formighieri and Melis [39] over-expressed an optimized *NaGLS* (geranialool synthase) in *Synechocystis*, resulting in a geranialool yield of 360 µg/g of dry weight. Rodrigues and Lindberg [29] demonstrated that the codon-optimized, bisabolene synthase (AgB) from *Abies grandis* significantly increased the production of (E)-α-bisabolene in *Synechocystis* 6803. However, the over-expression of the *ispA* gene encoding farnesyl pyrophosphate synthase from *E. coli* had no effect on bisabolene production. These studies underscore the importance of terpenoid synthases in terpenoid biosynthesis.

Furthermore, enzymes from diverse organisms exhibit distinct enzymatic activities and substrate preferences, which lead to varied yields of terpenoids in cyanobacteria. Consequently, enzymes with high enzymatic activity levels were explored and employed. For example, Gao et al. [34] over-expressed *ispA* from different organisms in *Synechococcus* 7942; the results showed that *ispA* from *Eucalyptus globulus* led to the highest isoprene yield, which was 25-fold greater than the lowest. Likewise, upon the expression of limonene synthases (LSs) from *Mentha spicata* and *Citrus limon* in *Synechocystis* 6803, the production value of limonene from *MsLS* was double that from *ClLS* [40]. However, by-product formation during terpenoid biosynthesis is a major concern as it competes for precursors, and consequently reduces the target terpenoid yield. Thus, inhibiting the by-product pathways is also an important strategy for enhancing terpenoid production. For example, the deletion of squalene hopene cyclase (Shc) in *Synechocystis* 6803, which blocks the conversion of squalene to hopene, along with the over-expression of the squalene synthase, resulted in a peak squalene production of 13.72 mg/L [41].

However, even when the appropriate genes are selected, the desired outcomes may not always be achieved, often due to the low gene level under strict regulation. The gene level can be precisely regulated both at the translational level by ribosome binding site (RBS) modification and at the transcriptional level through promoter optimization. In terpenoid biosynthesis research, it is common to optimize and over-express multiple genes simultaneously separated by RBSs under individual promoters. For example, in *Synechococcus* 7942, the production of limonene was increased from 8.5 to 32.8 µg/L/OD/d by employing an RBS [33]. In contrast, when the FPP synthase was conjugated with anthracene synthase by different RBSs in cyanobacteria, the anthracene titers showed significant variations ranging from undetectable to 7.9 mg/L [42]. Additionally, Zhong et al. [7] compared the functions of two distinct RBSs in *Synechocystis* 6803 for taxol precursor biosynthesis and demonstrated that the expression level and the connection order of RBSs can cause substantial differences in the final yield. These findings underscore the critical role of RBS selection in modulating the gene levels in cyanobacteria and highlight the importance of the appropriate RBSs for terpenoid synthesis.

In addition to RBS optimization, the production of engineered strains can also be limited by promoters. To enhance the expression of terpenoid synthases in cyanobacteria, strong promoters, such as P_cpc560_, P_psbA_, and P_trc_, are commonly employed [7,33]. However, the final yields are still significantly different. For example, Wang et al. [33] observed that replacing P_trc_ with P_sbA_ led to an over 100-fold increase in limonene production in *Synechococcus* 7942. Similarly, Zhong et al. [7] found that the expression of taxadiene-5α-hydroxylase in *Synechocystis* 6803 was higher under P_cpc560_ compared to P_psbA_, resulting in the elevation of total oxygenated taxanes production. Another noteworthy study also demonstrated the importance of promoter selection. At a normal temperature (28 °C), even when the 4,4’-β -carotene oxygenase (*CrtW*) and the 3,3’-β -carotene hydroxylase (*CrtZ*) genes were driven by a strong promoter in *Synechocystis* 6803, it was difficult to obtain the target product astaxanthin (Asx). However, when the strong promoter was replaced with a temperature-inducible promoter, a relatively high yield of Asx was successfully achieved [43]. These studies collectively highlight the critical role of promoter strength in adjusting terpenoid synthase expression and suggest that screening for suitable promoters is essential for enhancing terpenoid production in engineered cyanobacteria.

### 3.4. Optimization of Fermentation Conditions

Terpenoid biosynthesis in cyanobacteria is not only governed by metabolic pathways, but also by the culture conditions. Although most research is still conducted in a laboratory, there is growing interest in scaling it up to industrial levels [44]. To facilitate the efficient biosynthesis of terpenoids, the major strategies focus on modifying CO_2_ availability, light intensity, and nutrient composition [45,46,47].

High concentrations of CO_2_ (5–20%) can promote photosynthesis, and consequentially increase terpenoid production in cyanobacteria [48]. Thus, enhancing the photosynthetic efficiency of cyanobacteria is a key strategy for improving terpenoid yields. The high-density cultivation (HDC) system is particularly suitable for cyanobacterial cultivation, as it employs membrane-mediated CO_2_ delivery techniques combined with optimized nutrient supply and high light intensity (Figure 2). This system facilitates rapid and sustainable biomass accumulation [49,50]. Numerous studies have proved that the HDC system significantly enhances the yield of natural products in cyanobacteria [7,30]. For example, a *Synechocystis* 6803 strain showed substantial increase in the titer of (E)-α-bisabolene when cultured in the HDC system [29]. Similarly, the titer of taxadiene in *Synechocystis* 6803 elevated by 2–5 times when grown in the HDC system [7].

Light intensity and duration are also important for terpenoid biosynthesis. Englund et al. [32] engineered *Synechocystis* to produce manoyl oxide diterpenoids. Interestingly, the yield of diterpenoids was augmented by 4.2 times under low-light-level conditions (20 μmol photons m^−2^ s^−1^), whereas it remained unchanged under high-light-level conditions (100 μmol photons m^−2^ s^−1^). Other studies reported that continuous light results in higher metabolite titers than alternating light/dark cycles (12:12) [42,51]. Therefore, the modulation of CO_2_ concentrations, light intensity, and the photoperiod could further enhance terpenoid synthesis in cyanobacteria.

Given that cyanobacteria are capable of utilizing light energy for inorganic carbon fixation, other nutrients are essential for their growth and terpenoid production [25]. Nitrogen and phosphorus are particularly crucial for cyanobacterial metabolism. Although *Synechocystis* 6803 cannot fix atmospheric nitrogen because of lacking nitrogenase, it can store ammonia nitrogen in the form of cyanophycin [52]. However, nitrogen starvation can still lead to pigment degradation, chlorosis, and enzyme activity impairment, ultimately limiting photosynthetic carbon sequestration [53]. Similarly, although *Synechocystis* 6803 can store inorganic phosphate, phosphate starvation can still diminish photosynthesis and cause etiolation [54]. Therefore, maintaining sufficient nutrient levels during fermentation, including nitrogen and phosphorus, can avoid nutrient deficiencies and enhance the terpenoid yield in cyanobacteria. Therefore, by optimizing the light conditions and ensuring the supply of key nutrients, combined with the energy-saving and efficient cycle system of HDC, the yield of cyanobacteria terpenoids can be effectively increased. Although it is necessary to increase the input of some light energy and nutrients, the HDC system has significant economic advantages for the large-scale production of high-value-added terpenoids.

## 4. Future Perspectives

In recent years, with the rapid advancements in synthetic biology, cyanobacteria have emerged as a promising platform for the sustainable production of terpenoids. A variety of terpenoids, including bisabolene, limonene, forskolin, taxadiene, squalene, carotenoids, and astaxanthin, have been successfully synthesized in cyanobacterial systems, demonstrating their potential as renewable chassis of high-value compounds with the advantages of photoautotrophy. Many researchers have employed numerous strategies to enhance the terpenoid production in cyanobacteria, including the construction and optimization of metabolic pathways, the mining and modification of high-activity-level key enzymes, the utilization of optimal promoters, the optimization of RBSs, and the development of efficient fermentation systems. These strategies are generally applicable to various terpenoids and are often combined to maximize the yield of terpenoids. Despite the environmental and economic advantages of cyanobacteria, challenges still remain. The widely used model cyanobacteria strains exhibit slower growth, reproduction, and metabolism compared to those of heterotrophic microorganisms, such as *E. coli* and *S. cerevisiae,* leading to the lower terpenoid yields. These limitations are mainly caused by the low utilization efficiency of light energy and CO_2_. In addition, an insufficient supply of precursors for terpene synthesis in cyanobacteria, attributed to limited flux within the MEP pathway, as well as competition for carbon flow from photosynthetic by-products also serve as significant limiting factors.

To fully explore cyanobacteria as a sustainable platform for terpenoid production, these issues, including low conversion efficiency, slow growth rates, and the poor utilization of light and CO_2_, must be completely addressed. The following areas should be focused on in the future. First, the discovery of optimal cyanobacteria species with a high growth rate and conversion efficiency is critical for improving the final titer. In addition to well-studied Synechocystis 6803 and *Synechococcus* 7942, new strains with desirable characteristics have recently been reported. *Synechococcus elongatus* UTEX 2973, which shares 99.8% genomic similarity with *Synechococcus* 7942, exhibits an exceptionally short doubling time of 1.5 h, which is comparable to that of yeast. Furthermore, it can thrive under more extreme conditions (a high temperature of 42 °C and light intensity of 1500 μmol m^−2^ s^−1^) and achieve a maximum dry cell weight (DCW) biomass of 23.4 g/L in a semi-continuous culture, which is far beyond the reach of many model cyanobacteria [55,56,57,58]. Likewise, *Synechococcus* sp. PCC 11901 can endure temperatures up to 43 °C and light intensities of 600 μmol m^2^ s^−1^, with a doubling time of just 2.1 h and a remarkable DCW of 32.6 g/L [59]. These strains highlight the potential for further improvements in cyanobacterial chassis for industrial applications. Second, optimizing the terpenoid biosynthetic pathway is crucial for improving yields. The key strategies include RBS regulation, screening for optimal promoters, pathway modularization, high-activity-level enzymes selection, and enzyme mutagenesis. These approaches can help to streamline and enhance terpenoid synthesis in cyanobacteria. Third, establishing co-culture systems that combine cyanobacteria with heterotrophic microorganisms may offer a promising avenue for improving terpenoid production. The recent studies have introduced glucose-producing *S. elongatus* mutants into mitochondria-deficient yeast strains; and stable yeast–cyanobacteria chimeras that can grow on bicarbonate alone were isolated under selective pressure; and following the construction and transformation of an LS (from *Citrus × limon*) expressing a vector in the chimera, the presence of limonene was successfully detected using GC-MS analysis [60]. In summary, though significant progress has been made, the further exploration of the aforementioned and other strategies is urgently needed to improve terpenoid biosynthesis in cyanobacteria. We believe that the genetic modification of cyanobacteria has the potential to revolutionize photoautotrophy chassis and pave the way for the green and efficient synthesis of valuable terpenoids.

## Figures and Tables

**Figure 2 microorganisms-13-01342-f002:**
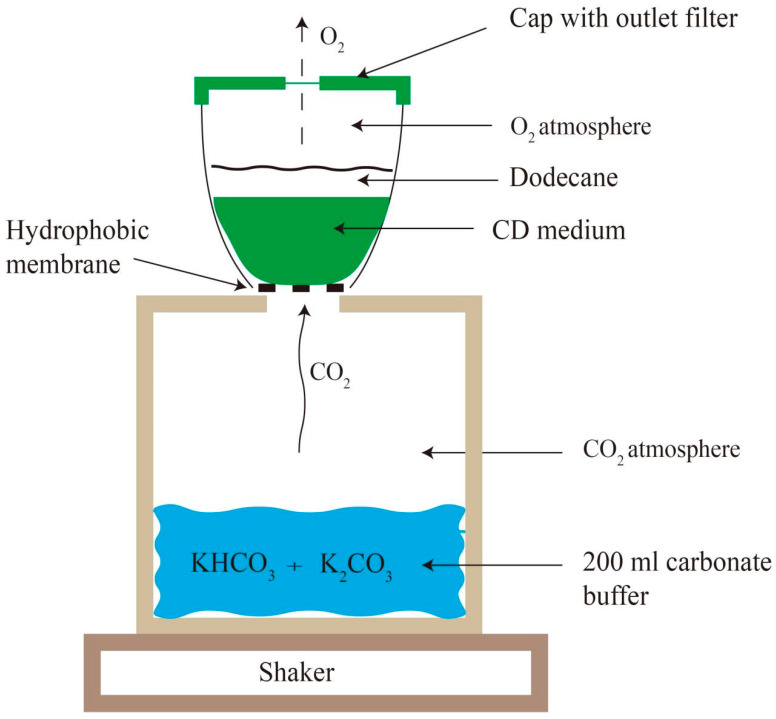
A schematic diagram of the HDC system. The highly concentrated bicarbonate/carbonate buffer constantly provides gaseous CO_2_ that diffuses through the hydrophobic membranes into the turbulent cell suspensions (in nutrient-rich CD medium). Oxygen (O_2_) is released from the cultivation vessels through the outlet filter in the cap. The culture is covered with 25% *v*/*v* dodecane and shaken in an environment of constant light and temperature. This schematic is adapted from [7].

**Table 1 microorganisms-13-01342-t001:** Recent studies on production of terpenoids in cyanobacteria. CpcB, β-subunit of phycocyanin; IspS, isoprene synthase; IspA, farnesyl phosphate synthase; TASY, taxadiene synthase; PHLS, phellandrene synthase; NptI, neomycin phosphotransferase I; Ps, patchoulol synthase.

Product	Strain	Titer	Time (Days)	Engineering Strategies	References
Isoprene	Synechocystis 6803	12.3 mg/g DCW	4	CpcB-IspS	[26]
Isoprene	Synechococcus 7942	105.2 mg/L	6	Ru5P-PP, IspG-IspH	[27]
Pinene	Synechococcus 7942	5.2 mg/L	6	Ru5P-PP + IspG-IspH, LS-GPPS	[28]
Limonene	Synechococcus 7942	21 mg/L	5	LS-GGPPS	[18]
β-Phellandrene	Synechocystis 6803	24 mg/g DCW	2	Exogenous MVA pathway, CpcB-PHLS, Nptl-GPPS	[18]
Taxadiene	Synechocystis 6803	2.94 mg/L	7	GGPPS-TASY, DXS-IspA	[7]
Bisabolene	Synechocystis 6803	186 mg/L	12	AgB-IspA, DXS-IDI	[29]
Patchoulol	Synechocystis 6803	17.3 mg/L	8	P_petE_-Ps	[30]
β-Caryophyllene	Synechococcus elongatus UTEX 2973	212.37 μg/L	6	IDI1-GPPS-IspA-TPS21	[31]

## Data Availability

The original contributions presented in this study are included in the article. Further inquiries can be directed to the corresponding author.

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
