# Peer review of "The Sustainable Production of Terpenoids in Cyanobacterial Chassis"

_microorganisms, 2025, doi:10.3390/microorganisms13061342_

Round 1
Reviewer 1 Report
Comments and Suggestions for Authors
Some sections of the manuscript require grammar and spelling corrections.
How did the authors select the information for this review manuscript? What were the criteria for including information in the manuscript?
The aim of the study is not clear.
Is this topic of relevance? Who might be the target audience for this work? Make sure to clarify this information in the introduction.
Figure legend 1 requires the authors to clarify whether they created the image.
What is the purpose of section 1? This information is too general.
Table 1 should include the production yield for each terpenoid.
The chemical structures of the terpenoids shown in Table 1 should be shown in another figure.
Do any of these terpenoids have pharmacological effects or any other applications? The manuscript should include a new section.
Do cyanobacteria present any advantage over E. coli and S. cerevisiae in the metabolite production? The manuscript should include this information.
The format of some references is not homogeneous (e.g., 24, 31, and 54).
Comments on the Quality of English LanguageSome sections of the manuscript require grammar and spelling corrections.
Reviewer 2 Report
Comments and Suggestions for Authors
The review dealing with the approaches of artificial biosynthesis of terpenoids has a nice general approach but lacks more concisive points to become publishable.
The main problem I have with the review is figure 1. It suggests a lot of different pathways and confuses a lot. Having compounds like paclitaxel which is at least a mixture of phenylpropanoid and terpenoid just as a result of terpene biosynthesis is misleading and planly wrong. The figure has to be carefully reworked to only contain terpenoids.
The figure caption of 1 is a miss. If you need half a page to describe all abbrevations it is maybe not a good fit. The figure caption should clearly express a message and that is sadly missing. Since figure 1 is one of the major parts of the manuscript the figure has to be completly reworked.
The same is for table 1, it is sadly very confusing too many abbrevations and common units should be used.
Reviewer 3 Report
Comments and Suggestions for Authors
The manuscript titled the sustainable production of terpenoids in cyanobacterial chassis by Hong et al presents a comprehensive review of metabolic‑engineering strategies to harness cyanobacteria for photoautotrophic terpenoid biosynthesis. The authors systematically cover both upstream (MEP/MVA) and downstream modules, key enzyme optimizations, heterologous pathway introduction, and bioprocess considerations. The manuscript is well written, up‑to‑date, and will serve as a valuable resource for researchers. Thus, I recommend acceptance after minor revision according to the comments below that will further enhance its clarity and impact.
Comments to the Authors:
- Section 2.4 would benefit from the addition of a brief techno economic perspective to help readers assess industrial feasibility.
- Section 3 should present a more balanced outlook. I suggest including a discussion of current limitations.
- Minor language and formatting edits
- In the Abstract, replace the semicolon after “Terpenoids” with a comma (Terpenoids, which are widely distributed…).
- In Section 1.1, line 12; “catalyze by seven enzymes” should read “catalyzed by seven enzymes”.
- Ensure consistency in enzyme nomenclature (e.g., sometimes “DXP synthase (DXS)” appears as “DXP-synthase (DXS)”); unify to one style.
- Legend of figure 1; please define all acronyms (e.g., AACT) in the legend rather than referring back to the text.
Round 2
Reviewer 1 Report
Comments and Suggestions for Authors
Can the authors specify which changes in grammar and spelling they performed?
How did the authors select the information for this review manuscript? What were the criteria for including information in the manuscript? The manuscript should include the reply provided by the authors.
What is the purpose of section 1? This information is too general. What modification did the authors perform in section 1?
The chemical structures of the terpenoids shown in Table 1 should be shown in another figure. I could not find the chemical structures of the compounds in the manuscript.
Do cyanobacteria present any advantage over E. coli and S. cerevisiae in metabolite production? The authors explained limited information in the introduction section about my recommendation.
Round 3
Reviewer 1 Report
Comments and Suggestions for Authors
The manuscript can be accepted for publication